# Vacuum Impregnation Process Optimization for Tilapia with Biopreservatives at Ice Temperature

**DOI:** 10.3390/foods11162458

**Published:** 2022-08-15

**Authors:** Yan Liu, Min Li, Zhi Jin, Jing Luo, Biao Ye, Jianwen Ruan

**Affiliations:** 1College of Mechanical and Power Engineering, Guangdong Ocean University, Zhanjiang 524088, China; 2College of Food Science and Technology, Guangdong Ocean University, Zhanjiang 524088, China; 3Guangdong Provincial Key Laboratory of Aquatic Product Processing and Safety, Guangdong Ocean University, Zhanjiang 524088, China

**Keywords:** vacuum impregnation, biopresavitives, tilapia, ice temperature, RSM

## Abstract

The vacuum impregnation (VI) process was used to pretreat tilapia fillets with biopreservatives at −2 °C. Response surface methodology (RSM) was utilised to optimize processing conditions, including vacuum pressure (p_v_), vacuum maintenance time (t_1_), and atmospheric pressure recovery time (t_2_), which were determined to be 67.73 kPa, 23.66 min, and 8.87 min, respectively. The anticipated values for the aerobic plate count (APC), total volatile basic nitrogen (TVB-N), and comprehensive score (CS) were 5.17 lg CFU/g, 14.04 mg/100 g, and 0.98, respectively. Verification experiments were conducted, and the experimental results for APC and TVB-N deviated from the predicted values by 0.19% and 0.64%, respectively. After 30 days of storage following VI and atmosphere impregnation (AI) pretreatment, the water-holding capacity (WHC), APC, TVB-N, hardness, and whiteness were determined. On the 30th day, the results for VI pretreatment were 63.38%, 6.27 lg CFU/g, 17.41 mg/100 g, 3.11 N, and 47.73, respectively. Compared with AI pretreatment, WHC, hardness, and whiteness increased by 14.8%, 18.6%, and 6.3%, respectively, whereas APC and TVB-N decreased by 11.3% and 29.6%, respectively. This study demonstrates that when biopreservatives are applied during the pretreatment process, VI technology can be utilised to facilitate their penetration into the interior of tilapia, hence significantly enhancing the effect of ice-temperature preservation.

## 1. Introduction

China is the largest producer of tilapia in the world. Production is anticipated to reach 13.25 million tonnes by 2025 [1]. Tilapia is a high-protein perishable fish. Improper storage increases the total amount of bacteria in the product, and protein degradation generates volatile basic nitrogen, which alters the water-holding capacity, texture, and colour of the fish. Consequently, tilapia product preservation technology is essential for processing, transport, storage, and marketing. Ice-temperature storage is common practice for preserving the quality of aquatic products during storage and distribution. Because the hardness, colour, water-holding capacity, and other characteristics of seafood are better in ice-temperature samples than in frozen storage, consumers prefer ice temperature storage products more [2,3]. Usually, a sufficient quantity of biopreservatives is added to extend the storage time at ice temperature. Compared to chemical preservatives, biopreservatives are benign, non-toxic, and edible, which is why customers choose biopreserved products.

For a long time, impregnation was the most prevalent method of food preservation. However, because the impregnation process takes longer, preservatives remain on the surface and are unable to penetrate the interior of the food tissue; hence, the expected preservation conditions are not satisfied.

As a result, various researchers have investigated increased impregnation processes, with vacuum impregnation (VI) being the most prevalent. The VI process includes submerging the food in a solution containing a specified concentration of preservatives or brine. Then, vacuum pressure (p_v_) is applied to the immersion solution, maintained for a certain time interval (t_1_), and then atmospheric pressure is restored and maintained for a predetermined time duration (t_2_). The VI process deforms and relaxes food tissue and intercellular space while also introducing a hydrodynamic mechanism (HDM) that affects not only the dynamic and equilibrium states of the system, but also the physical, mechanical, and microstructural properties of the food structure, all of which contribute to the enhancement of solute diffusion processes within food tissues [4,5].

VI is commonly used to preserve the colour and prevent browning in fruits and vegetables during processing and storage [6,7,8], in the addition to functional additives such as fortified calcium [9,10,11,12,13], probiotics [14,15], and iron [16,17], as well as in the saline treatment of aquatic products [4,18,19] and meat [20,21,22].

However, little study has been conducted on the use of biopreservatives in combination with VI technology to improve the quality of aquatic goods preserved at ice temperature. X. Zhao et al. [23] employed VI to pretreat seabass fillets in a solution of fish gelatin and grape seed extract at 4 °C. Compared to AI, VI significantly increased the WHC, decreased the total number of bacterial colonies, and reduced biogenic amines. Y. Zhang et al. [24] prepared and tested bone hydrolysates from bighead carp (*Aristichthys nobilis*), and the hydrolysates with highest antioxidant activity were utilised to pretreat bighead carp with VI. Compared to controls, the vacuum-impregnated fillets had higher sulfhydryl contents, lower thiobarbituric acid-reactive substances, superior texture, and greater water-holding capacity after freeze–thaw cycle. Shiekh Khursheed et al. [25,26] used VI, pulsed electric field, Chamuang leaf extract and high-voltage cold plasma to preserve Pacific white shrimp (*Litopenaeus vannamei)* at 4 °C. The experimental groups showed lower lipid oxidation, bacterial counts, total volatile bases, and protein carbonyls than the control groups. X. Zhao and colleagues [27,28] discovered that using VI to aid fish gelatin and grape seed extract for preserving tilapia fillets resulted in a significant increase in fish quality and a decrease in microbial survival. M. Li et al. [29] applied ultrasonic assisted with VI and pulsed VI in trehalose solution to investigate the effects on water migration and volatile components of heat pump-dried tilapia fillets. They discovered that the samples pretreated with the pulsed VI had better protein protection features.

The aforementioned researchers, as well as those studying the salting of fish and meat [5,18,19,21], have focused on the effect of VI on food quality. The important parameters, p_v_, t_1_, and t_2_, used in VI technology are arranged in Table 1 of those publications. It is evident that VI parameters vary substantially amongst sources and are irregular, and nearly all studies were conducted above 0 °C. According to the operating mechanism of VI technology, the mass transfer effect during the impregnation process is influenced by p_v_, t_1_, and t_2_; therefore, these three parameters are very important. Currently, no study has optimised the VI parameters utilized in the VI method for tilapia maintained at ice temperature.

In a previous study, the research team optimised a formulation of biopreservatives (sodium alginate 8 g/L, Nisin 0.8 g/L, and sodium erythorbate 7.5 g/L, in mass concentrations) for the preservation of tilapia at ice temperature (−2 °C) [36]. Compared to the control, the shelf life of tilapia fillets treated with the biopreservatives were extended to 22 days. In this study, VI technology was used to improve the fresh-keeping effect and to investigate the effects of the VI process parameters p_v_, t_1_, and t_2_ on the quality of tilapia fillets. Using response surface methodology (RSM), the study intended to optimise the VI process parameters. Additionally, experiments were conducted with the optimized parameters to determine the influence of biopreservatives’ supplemental VI technology on the quality of tilapia stored at −2 °C.

## 2. Materials and Methods

### 2.1. Materials, Reagents, and Equipment

Tilapia (*Oreochromis niloticus*) was purchased at the Huguang Market in Mazhang District of Zhanjiang City, Guangdong Province, China. This experiment employed only food-grade reagents. Sodium alginate was obtained from Yuanye Biotechnology Co., Ltd., Shanghai, China. Nisin was acquired from Wanlida Biotechnology Co., Ltd., Hebei, China. Sodium erythorbate was purchased from Liyuan Food Additives Co., Ltd., Guangzhou, China. 2-thiobarbituric acid was obtained from Sinopharm Chemical Reagent Co., Ltd., Shanghai, China; plate counting agar was obtained from Land Bridge Technology Co., Ltd., Beijing, China.

Vacuum apparatus (VCD-02), Shanghai Xianlv Vacuum Fresh-keeping Equipment Co., Ltd., Zhejiang, China; Analytical Balance, Shimadzu Instrument (AUY220) Co., Ltd., Tokyo, Japan; vacuum packaging machine (DZ400/2D), Reili Packaging Machinery Co., Ltd. Zhejiang, China; Thunder Magnetic PH Meter (PHS-3C), INESA Scientific Instrument Co., Ltd., Shanghai, China; homogenizer (125), Shanghai Eiken Machinery Equipment Co., Ltd., Shanghai, China; HHS constant temperature water bath, Boxun Industrial Co., Ltd., Shanghai, China; Kjeldahl nitrogen analyzer (Vap450), Gerhardt Co., Ltd., Bonn, Germany; electric heating constant temperature incubator (HPX-9082MBE), Shanghai Boxun Industrial Co., Ltd., Shanghai, China; double single-sided purification workbench (SW-CJ-2D), Suzhou Purification Equipment Co., Ltd., Suzhou, China; vertical pressure steam sterilizer (LDZX-50KBS), Shanghai Shen’an Medical Equipment Factory, Shanghai, China; texture analyzer (TMS-Pro), FTC, Washington D.C., USA; colorimeter (CR-10), Konica Minolta, Tokyo, Japan.

### 2.2. Preparation of Tilapia Fillets

After purchasing live tilapia, approximately 0.9 kg each, they were pre-cultured in a laboratory tank at a room temperature of 24 °C. In preparation for conducting the studies, the head, bone, and skin of the tilapia were removed, and the meat on both sides of the back was sliced into approximately 1 cm-thick fillets with a stainless steel knife. The fillets were cut into 12 cm × 5 cm × 1 cm rectangles and weighed around 80 ± 2 g.

The fish fillets were placed in beakers containing preservative liquid (25%, *w*/*w*) produced with sterile water [36]. The beakers were then placed within the vacuum chamber of the vacuum apparatus. Experiments were conducted in triplicate.

### 2.3. Impregnation Weight Gain Rate, WGR

At the end of the impregnation period, the fish fillets were taken out, placed in a sterile environment, drained until there were no remaining dops, and weighed. After absorbing the surface moisture of the fish fillets with sterile filter sheets, the fillets were weighed again. The weight growth rate of the impregnated fish fillets is calculated using Equation (1):(1)$$\mathrm{WGR}=\frac{m_{2}-m_{1}}{m_{1}}\times 100\%$$
where WGR is the impregnation weight gain rate, %; m_1_ is the mass of the fish fillet before treatment, g; m_2_ is the mass of the fish fillet at the end of treatment, g.

### 2.4. TVB-N

TVB-N concentration was determined using a Kjeldahl nitrogen analyzer in accordance with the “Chinese National Food Safety Standard, Determination of Volatile Base Nitrogen in Food, GB/5009.228-2016”.

### 2.5. Aerobic Plate Count, APC

The total number of bacterial colonies was determined using the aerobic plate count method during storage. The APC was determined following the procedure outlined in “Chinese National Food Safety Standard, Food microbiology test—Determination of aerobic plate count, GB/4789.2-2016”. The expressed amount of APC was in log colony-forming units (CFU).

### 2.6. Water-Holding Capacity, WHC

According to Lakshmanan’s method [37], weigh 2 g of fish meat accurately, wrap it in two layers of known-weight filter paper, and place it in a centrifuge tube. After centrifuging at 10 °C and 3000 rpm for 10 min, the filter paper was weighed. WHC was calculated according to Equation (2):(2)$$\mathrm{WHC}=\left(1-\frac{m_{2}-m_{1}}{m_{0}}\right)\times 100\%$$
where m_0_ is the mass of the tilapia meat, g; m_1_ is the weight of the filter paper before centrifugation, g; m_2_ is the weight of the filter paper after centrifugation, g.

### 2.7. Hardness

The samples were evaluated using the texture analyzer’s texture profile analysis (TPA) mode. The diameter flat-bottomed column probe P/5 was used, the induction force was set to 1000 N, the test speed was set to 60 mm/min, the starting force was set to 0.5 N, the deformation was set to 50%, and the interval between two depressions was set to 1 s. Five measurement were taken of each piece of fish, and the average was calculated.

### 2.8. Whiteness

The value of whiteness can be used to determine the colour of the fish fillets. The colorimeter was used to determine the L* value (lightness on a scale of 0 to 100 from black to white), the a* value (+for red or −for green), and the b* value (+for yellow or −for blue) of the fish fillets [38]. The value of whiteness is calculated using Equation (3):(3)$$W=100-\sqrt{{\left(100-L^{*}\right)}^{2}+a^{*}{}^{2}+b^{*}{}^{2}}$$
where W represents the whiteness index; L* is lightness, a* is the red-green value, and b* is the yellow-blue value.

### 2.9. Experimental Design and Statistical Analysis

#### 2.9.1. Single-Factor Experiments

To investigate the effect of a single parameter on the experimental results, single-factor experiments using the control variables method were conducted, which means that only one variable was used to represent the influencing factors on the experimental results, and all other variables remained constant [39]. This section investigates the effect of p_v_, t_1_, and t_2_ on the quality index of tilapia fillets for WGR, APC, and TVB-N.

Each time, five groups of prepared fish fillet samples (total 150 fillets) were processed for single-factor tests. Based on previous experiments, fist, t_1_, and t_2_ were fixed at 15 and 6 min, respectively, and p_v_ was changed to 0 (atmospheric pressure), 20, 40, 60, and 80 kPa. Then, p_v_ and t_2_ were held constant at 60 kPa and 6 min, respectively, while t_1_ was adjusted to 5, 10, 15, 20, and 25 min. Finally, p_v_ and t_1_ were set to 60 kPa and 15 min, respectively, and t_2_ was varied to 3, 6, 9, 12, and 15 min. Following the preceding procedures, the fillets were placed in a sterile environment to drain until no water remained. The drained fish fillets were subsequently vacuum-packed and stored at −2 °C. The indicators of WGR, APC, and TVB-N were measured every 5 days in all experimental groups.

#### 2.9.2. RSM Experiments

RSM was used to optimise the VI process for tilapia fillets. The Box–Behnken design (Table 2) was employed, because this investigation required three levels with equal spacing and fewer runs [32,40]. The RSM experiment was created, analyzed, optimized, and plotted by Design Expert software (Version 8.0, Stat-Ease, Inc., Minneapolis, Minnesota, USA). On day 25, two response parameters were evaluated experimentally: APC and TVB-N. Through calculation, a single response, the comprehensive score (CS), was simultaneously determined. The statistics are summarized in Table 3. Using regression analysis [41], second-order polynomial equations were fitted to the responses in terms of the independent variables.

Equation (4) is a quadratic model that was applied to characterise the response variables. This model describes the response variables Y (APC, TVB-N, and CS) as a function of the factor variables X_i_ (i assuming values from 1 to 3) for all applicable VI solutions:(4)$$Y=a_{0}+\sum^{\;}a_{i}X_{i}+\sum^{\;}a_{\mathrm{ii}}X_{i}^{2}+\sum^{\;}a_{\mathrm{ij}}X_{i}X_{j}$$

In this study, Equation (5) was derived for the solutions based on Equation (4),
Y = a_0_ + a_1_p_v_ + a_2_t_1_ + a_3_t_3_ + a_12_p_v_t_1_ + a_13_p_v_t_2_ + a_23_t_1_t_2_ + a_11_p_v_^2^ + a_22_t_1_^2^ + a_33_t_2_^2^(5)
where a_0_ represents the constant, a_i_ represents the linear effect, a_ii_ represents the quadratic effect, and a_ij_ represents the factor interaction effect. Consequently, the main effect coefficients are a_1_, a_2_, and a_3_, the quadratic main effect coefficients are a_11_, a_22_, and a_33_, and the two factor interaction coefficients are a_12_, a_13_, and a_23_. Positive or negative coefficient values in Equation (4) imply a reaction-promoting or reaction-inhibiting influence, respectively. Following the development of polynomial models, analysis of variance (ANOVA) was used to examine how well the model described the data at a 95% confidence level, and *p*-value criteria were applied to establish statistical significance. R^2^ is the model’s coefficient of determination, and the lack of fit is used to evaluate the model’s precision. The statistical significance of the coefficients was determined using the software’s built-in ANOVA tool.

To initiate the optimization process, a composite score was calculated for APC and TVB-N using the Membership comprehensive scoring method, with a lower value indicating superior performance. This is because the smaller the aforementioned statistics were, the higher the storage quality of the fillets. According to Equation(6), the APC and TVB-N memberships were weighted independently:(6)$$l_{i}=\frac{c_{\mathrm{imax}}-c_{i}}{c_{\mathrm{imax}}-c_{\mathrm{imin}}}$$
where c_i_ is the index value, c_imin_ is the minimum value of the index, c_imax_ is the maximum value of the index, and I = 1 or 2, where 1 stands for APC, and 2 stands for TVB-N.

Comprehensive scores (CS) were calculated with Equation (7):CS = al_1_ + bl_2_(7)
where l_1_ represents the Membership of APC, l_2_ represents the Membership of TVB-N, coefficient a represents the weight of APC, and coefficient b represents the weight of TVB-N. Given that both metrics have an equal impact on the fish fillets’ quality, a = b = 0.5. The CS should be as large as possible, according to the analysis of Equation (6).

The mean ± standard deviation of the results was calculated. The statistical significance was determined using SPSS (Version 18.0, SPSS Inc., Chicago, IL, USA) statistical analysis software for the one-way analysis of variance (ANOVA) with Duncan’s test. The significance of differences was established at the 5% probability level (*p* < 0.05). The analysis figures of the experimental data were designed with Origin software (Version 8.0, OriginLab, Northampton, MA, USA).

## 3. Results and Discussion

### 3.1. Result and Discussion of Single-Factor Experiments

Table 4 depicts the impact of various p_v_ on the WGR, APC, and TVB-N. The WGR increased significantly (*p* < 0.05) when p_v_ was increased from 0 to 60 kPa; however, when p_v_ was increased from 60 kPa to 80 kPa, the WGR changed insignificantly (*p* > 0.05). The APC and TVB-N values for each fillet group increased as storage duration increased. The APC tended to initially decrease and subsequently increase as p_v_ increased over the same storage time. This tendency was at its lowest point when sustained for 10, 15, and 20 d at 60 kPa, but it increased when p_v_ approached 80 kPa. The TVB-N value declined as p_v_ increased on 5 d and 10 d, but on 15 d and 20 d, it decreased to a minimum at 60 kPa and then increased at 80 kPa. This demonstrated that p_v_ does not imply the higher the better. In this investigation, the optimal p_v_ was 60 kPa; when p_v_ was greater than 60 kPa, the storage quality of the tilapia fillets was no longer significantly improved, and it might even deteriorate. According to the research of J.W. Kang et al. [42], this is mostly likely because the biopreservative liquid has completely filled the fish cell spacing and reached a saturated state at 60 kPa, the internal and external pressures have reached equilibrium, and the biopreservatives can no longer penetrate the fish tissue. Therefore, increasing p_v_ above 60 kPa has no obvious impact on the preservation effect. Moreover, as p_v_ increased, the fish tissues might have been progressively damaged, resulting in a decrease in storage quality.

Table 5 demonstrates the effect of different t_1_ on the WGR, APC, and TVB-N. When p_v_ and t_2_ are held constant, the WGR value of the fish fillets steadily increased as t_1_ increased from 5 to 20 min, but became negligible after 15 min. When t_1_ was extended to 25 min, the WGR decreased. At the same storage time, as t_1_ increased, the APC and TVB-N values of each group decreased significantly from 5 to 20 min, but increased at 25 min, indicating that a longer impregnation period is not always preferable for tilapia fillets.

Table 6 depicts the influence of various t_2_ on the WGR, APC, and TVB-N. As can be seen, increasing t_2_ from 3 to 15 min had no discernible effect on WGR, although it tended to increase and then decrease. The WGR increased as t_2_ grew at and before 9 min, but there was a slight decrease at 12 and 15 min, indicating that t_2_ does not imply that a longer impregnation period is better. The APC and TVB-N decreased initially (3~9 min) and then increased (9~15 min) in response to an increase in t_2_ for the same storage duration. This may be because during the t_1_ stage, the cell structure of the fish fillets acquired a vacuum pressure, and then during the t_2_ stage, the system pressure was restored to atmospheric pressure. Obviously, the pressure in the t_2_ stage was greater than the pressure in the t_1_ stage. Under the pressure difference, the external biopreservation solution gradually permeated the interior of the fish fillets, and the permeation amount reached a saturation point at 9 min. Increasing t_2_ beyond 9 min had no significant effect on the biopreservatives’ fresh-keeping effect. This phenomenon is similar to Fito’s experiments with fruits [43], but fish meat is quite different from fruits, and the effect of t_2_ in this investigation was insignificant (*p* > 0.05).

According to the results of the single-factor experiments, the RSM experiment’s parameters, p_v_, t_1_, and t_2_, were selected as indicated in Table 2.

### 3.2. Result and Discussion of RSM Experiments

#### 3.2.1. Influence of Input Variables on Responses

Table 7 presents the linear, quadratic, and interaction effects of the independent variables on APC, TVB-N, and CS. In addition, Table 8 shows the regression coefficients of responses in terms of actual factors specified in the original units for each factor. The correlation coefficient R^2^ is significantly greater than 0.9, suggesting that the model fits the data well. In this investigation, the R^2^ values for APC, TVB-N, and CS were reported to be 0.955, 0.934, and 0.955, respectively, indicating that the model significantly matched the experimental data. Nearly all quadratic factors exhibited highly significant effects (*p* < 0.01) on response variables. Among the linear factors, t_1_ had a significant impact on all of the response parameters, whereas p_v_ had a significant effect on APC and CS, but an insignificant influence on TVB-N. The linear factor t_2_ and all interaction factors were determined to be insignificant on all response variables.

APC was significantly influenced by the linear factors p_v_ (*p* < 0.01) and t_1_ (*p* < 0.001), as well as the quadratic factors p_v_, t_1_, and t_2_ (*p* < 0.01). Because the quadratic regression coefficients of APC were all positive, when p_v_, t_1_, and t_2_ increased, the APC was observed to initially decrease and then increase; the three-dimensional response surface must be a surface with an upward opening (Figure 1).

TVB-N was significantly impacted by linear factor t_1_ (*p* < 0.01), and the quadratic factors p_v_ (*p* < 0.01), t_1_ (*p* < 0.05), and t_2_ (*p* < 0.01). The remaining factors were insignificant. It was revealed that as p_v_, t_1_, and t_2_ grew, TVB-N first decreased and then increased due to all the positive quadratic regression coefficients; the three-dimensional response surface must be a surface with an upward opening, too (Figure 2).

Computed from APC and TVB-N, which have identical weights for the effect on fish storage quality, CS was significantly influenced by linear terms of p_v_ (*p* < 0.05), t_1_ (*p* < 0.001), and all the quadratic factors (*p* < 0.01). In accordance with the mathematical relationship (Equations (6) and (7)), the smaller APC and TVB-N are, the higher the storage quality and the greater the CS. Furthermore, CS should be opposite to APC and TVB-N, as suggested by the results of the variance analysis (Table 8), which revealed that the quadratic terms were characterised by negative regression coefficients. Consequently, when the independent variables p_v_, t_1_, and t_2_ increased, CS initially increased and then decreased; the three-dimensional response surface must be a surface with a downward opening (Figure 3).

In conclusion, there must be a combination of p_v_, t_1_, and t_2_ that simultaneously minimises APC and TVB-N while maximising CS.

According to the F test (F value in Table 7), the influence sequence of the three independent variables on the three response parameters is consistent: t > p_v_ > t_2_, indicating that t_1_ had a larger influence than p_v_, and p_v_ has a greater effect than t_2_. Figure 1, Figure 2 and Figure 3 also support this conclusion. Therefore, if VI technology is used to enhance the effect of biopreservatives on the storage of tilapia at ice temperature, the appropriate vacuum duration should be chosen first, followed by the determination of the vacuum pressure, and the atmospheric pressure restore time has the least influence on storage quality.

#### 3.2.2. Optimisation of the Process Variables

Using a numerical optimization strategy based on the desirability approach and Design Expert 8.0 software, the optimal settings of the independent variables for VI technology were identified. The optimisation criterion for restrictions was to minimise APC and TVB-N while maximising CS. The independent variables for the optimization technique were kept within the experimental range. The optimised values for p_v_, t_1_, and t_2_ were 67.73 kPa, 23.66 min, and 8.87 min, respectively. The estimated APC, TVB-N, and CS values under ideal conditions were 5.17 lg CFU/g, 14.04 mg/100 g, and 0.98, respectively. To accommodate laboratory operations, experiments were conducted in triplicate with the adjusted parameters pv = 68 kPa, t1 = 23′40″, and t2 = 8′50″. The experimental values for APC and TVB-N were 5.17 ± 0.06 lg CFU/g and 14.23 ± 0.18 mg/100 g, respectively. CS was determined to be 0.96 compared to the predicted value of 0.98; the relative error is around 2.6%, indicating a satisfactory fit between the predicted and experimental values.

Pretreated with the biopreservatives, the storage quality of AI and optimal VI was compared for 30 days of storage at −2 °C. Every five days, indicators of APC, TVB-N, WHC, hardness, and whiteness were tested, and data were collected (Figure 4). The subsequent sections investigated the variation in these indicators.

TVB-N

The TVB-N value was applied as an indicator of fish spoilage [44,45], so it is an important indicator of fish storage quality. The majority of the increase in TVB-N resulted from the degradation of amino acids, protein, and some nitrogen-containing substances by spoilage bacteria and endogenous enzymes, resulting in the formation of volatile bases [28]. These enzymes resulted in the synthesis of fish-off-flavoring nitrogen compounds, including ammonia, monoethylamine, dimethylamine, and trimethylamine [46]. On day 25, the TVB-N value of AI exceeded the maximum limit standard for TVB-N in “China’s national food safety standards, fresh and frozen animal and aquatic products” (20 mg/100 g), whereas the TVB-N value of the fish fillets pretreated with VI remained within the standard range on day 30, but exceeded the limit standard on day 35. By day 35, the fish fillets had already begun to show deterioration; therefore, other quality-related indicators, APC, WHC, hardness, and whiteness were only detected for 30 days.

The TVB-N values of the fish fillets increased with storage time in both groups (Figure 4a), and these changes were almost identical to the changes in APC (Figure 4b). At the completion of the storage period, the TVB-N concentration increased from 9.79 mg/100 g to 24.71 mg/100 g (AI) and 17.41 mg/100 g (VI). With VI pretreatment, the TVB-N value did not increase significantly until day 15, especially during the first five days (*p* > 0.05). This demonstrates that VI treatment enhances the preservation effect of the biopreservatives, effectively extending the shelf life of tilapia fillets stored at ice temperature, because VI treatment effectively increases the penetration of preservatives into the fish tissue and inhibits protein oxidation. This result is comparable to the findings of X. Zhao et al. [28], who observed the lowest TVB-N value in tilapia fillets throughout storage in the combined pretreatment with fish gelatin and grape seed extract assisted by VI, compared to the control and the groups without VI. A. Andres-Bello et al. [30] studied the gilthead sea bream fillets pretreated with VI in solution containing lactic acid bacteria and nisin, respectively, and stored at 4 °C for 15 days; the TVB-N value assisting with VI was also lower than the control throughout the storage time. However, the TVB-N and other physico-chemical properties exhibited insignificant differences between the experiment groups; this could be because the VI process parameters used in the research were not large enough compared to the other researchers (Table 1).

APC

APC growth curves in fish fillets were fitted with the Baranyi equation [47]. With and without VI, the APC values increased gradually (Figure 4b). The average initial APC concentration of samples was 4.49 lg CFU/g. In the VI group, the APC rose more slowly and did not reach a significant level (*p* < 0.05) until day 15, particularly during the first 5 days (*p* > 0.05). On day 25, the APC of AI and VI pretreatment was 6.34 ± 0.25 lg CFU/g and 5.16 ± 0.10 lg CFU/g, respectively. The final APC of AI and VI pretreatment was 6.51 ± 0.16 lg CFU/g and 5.78 ± 0.21 lg CFU/g, respectively.

The hydrodynamic mechanism (HDM) [43] of VI may have led to a decrease in APC concentrations with VI pretreatment. This demonstrates that the auxiliary VI technology improves component penetration into the fish fillet and inhibits APC proliferation during the preservative impregnation pretreatment.

WHC

WHC refers to the capacity of fish to retain water during processing and storage. WHC is mostly related to the quantity of water that flows with difficulty, the protein gel, and the electrostatic charge of the fish. The higher the WHC, the stronger the binding ability of the fish network structure to water and other substances, indicating a denser spatial network structure in the gel [48,49]. Therefore, the WHC of fish meat is an important indicator of its preservation quality.

During storage, the multiplying bacteria in the fillets produce proteases that degrade the proteins and disrupt their gel structure, resulting in a reduction in WHC (Figure 4c). The initial WHC of both groups of fillets was 82.2%. During the storage period, the WHC of the tilapia fillets declined gradually in both pretreatment groups, but it decreased more dramatically in the AI pretreatment group, particularly in the first 10 days, decreasing to 63.80%. In the VI pretreatment group, the decline was slower, with no significant decrease in WHC over the first 5 days and a substantial decrease on day 10 to 72.21%. At day 30, the WHC of AI and VI pretreatment was 55.20% and 63.38%, respectively. This indicates that the adoption of VI technology in pretreatment reduced the degradation of the protein gel structure and maintained the amount of electrostatic charge, hence increasing the WHC of the fillets. Wang Z Y et al. [50] revealed that pulsed vacuum brining enhanced the WHC of lamb compared to atmosphere brining. Leal-Ramos et al. [20] determined that VI is an effective approach for increasing the moisture content of meat. This may be due to the fact that VI can expand muscle fibre to enhance the WHC [51].

Hardness

Hardness is considered the most essential textural property of fish [38]. The high-activity autolytic enzymes hydrolyze proteins and other connective tissues, thereby accelerating muscle deterioration [52] and diminishing tissue hardness. Dunajski et al. [53] reported that fish with higher moisture content had a softer texture, suggesting that water content can also affect the hardness of fish. In other words, the fish with lower water content had a harder texture. In this investigation, the hardness of fish samples from both groups decreased during the preservation period (Figure 4d). In the AI group, the hardness declined dramatically over the first 15 days, from 5.44 N to 2.48 N, whereas in the VI group, the hardness decreased significantly in the first 10 days, from 5.44 N to 3.68 N, and then slowed. At the same storage time, the hardness difference between the two groups was not statistically significant. But the hardness of the VI group was still greater than that of the AI group. The AI group’s lowest point of hardness occurred at 15 d (2.48 N), after which it increased slightly, whereas the VI group had a low point—though not its lowest—at 15 d (3.21 N). In the first 15 days, the effect of moisture loss on increasing hardness was much smaller than the effect of protein gel structure degradation on decreasing hardness, causing a rapid reduction in hardness. At 15 d, the degradation of the gel structure reached an extreme value, and the water loss continued to increase the hardness, resulting in a slight increase in hardness at 20 d. At 20 d, the water loss may have reached an extreme value, whereas the breakdown of the protein continued to decrease the hardness. The hardness did not change much after 20 d, but decreased gradually overall. The lowest hardness of VI pretreatment was observed at 30 d (3.11 N). This suggests that the VI technology can assist the biopreservatives in protecting the protein structure, delaying the deterioration of hardness and preserving the fillets’ aesthetic appearance. In contrast, increasing water loss during the VI dehydration of apples [54] and bamboo shoots [55] softened the tissue and reduced the hardness. This may be because fish have a completely different tissue structure than that of plants.

Whiteness

It has been demonstrated that colour change in fillets is highly correlated with storage quality [38], and whiteness is the most important colour indicator for white-meat fish such us tilapia, whose fresh meat is a beautiful white colour. Whiteness is the comprehensive value of L*, a*, and b* according to Equation (3), where L* is positively correlated with whiteness, and a* and b* are negatively related with whiteness. Therefore, whiteness decreases as the value of red, green, blue, or yellow increases. During ice temperature storage, the formation of ice crystals causes the rupture of muscle and fat cells, causing the infiltration of water and lipids onto the surface and the aggregation of myofibrillar proteins, which can enhance the reflection of light and increase brightness, thereby enhancing whiteness. However, the disruption of protein structures by enzymatic reaction and bacterial degradation leads to the oxidation of myoglobin to metmyoglobin, which has a reddish hue, and consequently reduces the whiteness. Moreover, the oxidation of lipids causes the aggregation of aldehydes and migration to the surface, resulting in a yellowish tinge that decreases the whiteness of the fillets [56,57,58,59].

In this study, the whiteness of VI was consistently greater than that of AI during the same period of storage (Figure 4e). Both groups reached their lowest value at 30 d, with 44.92 (AI) and 47.73 (VI). The initial whiteness of the fresh fillets was 52.73, and it decreased significantly over the course of the first 10 days in both groups, to 46.33 (AI) and 48.58 (VI). This indicated that during the initial period of storage, protein structural degradation and lipid oxidation predominated. After that, for AI pretreatment, the whiteness increased slightly at 15 d, and then decreased slowly, whereas for VI pretreatment, whiteness increased slightly until 20 d and subsequently declined slowly. Increasing whiteness during the middle of the storage period indicated that moisture migration was the dominant process at this stage. This transformation was similar to the change in hardness. The VI treatment facilitated effective penetration of the biopreservatives into the interior of the fillets, establishing a protective layer both inside and outside the fillets, slowing the process of protein denaturation, water loss, and fat oxidation, thereby preserving the colour of the fish fillets.

## 4. Conclusions

In this investigation, RSM was employed to optimise the process parameters of VI pretreatment of tilapia fillets in order to improve the preservation impact of biopreservatives stored at −2 °C. The following are our conclusions:

1. Using APC, TVB-N, and CS as response variables, the optimised VI process parameters obtained by RSM were p_v_ = 67.73 kPa, t_1_ = 23.66 min, and t_2_ = 8.87 min; according to ANOVA analysis, among the three parameters, the importance on quality effect was t_1_ > p_v_ > t_2_; that is, t_1_ had the greatest impact on the quality of tilapia, whereas t_2_ had the least impact.

2. The optimal parameters for verification experiments were adjusted to p_v_ = 68 kPa, t_1_ = 23′40″, and t_2_ = 8′50″. After 25 days of storage, the difference between the obtained and predicted CS value was 2.6%.

3. Using the optimised process parameters, tilapia fillets were pretreated with biopreservatives, and the quality indicators of tilapia fillets stored at −2 °C for 30 days were monitored. WHC, hardness, and whiteness increased by 14.8%, 18.6%, and 6.3%, respectively, in comparison to the AI effect, and APC and TVB-N decreased by 11.3% and 29.6%, respectively. The storage time for the tilapia was extended to 30 days. With the optimal parameters, it is obvious that VI technology can enhance the impregnation effect of biopreservatives, improve the storage quality of tilapia, and prolong the shelf life of tilapia at ice temperature.

## Figures and Tables

**Figure 1 foods-11-02458-f001:**
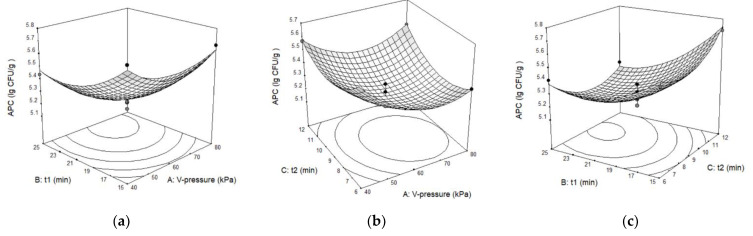
Three-dimensional response surface graphs for aerobic plate counts (APC) as a function of (**a**) p_v_ and t_1_. (**b**) p_v_ and t_2_ (**c**) t_1_ and t_2_.

**Figure 2 foods-11-02458-f002:**
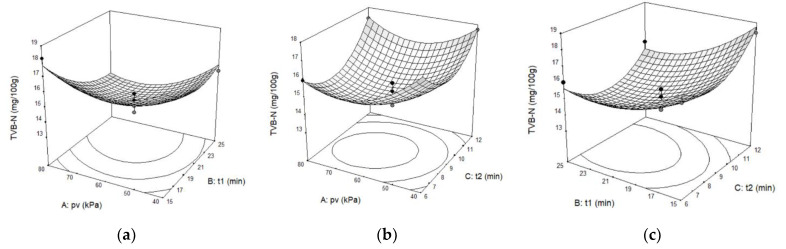
Three-dimensional response surface graphs for TVB-N values as a function of (**a**) p_v_ and t_1_. (**b**) p_v_ and t_2_. (**c**) t_1_ and t_2_.

**Figure 3 foods-11-02458-f003:**
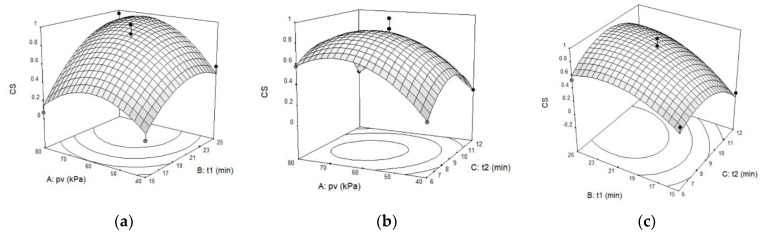
Three-dimensional response surface graphs for comprehensive score (CS) as a function of (**a**) p_v_ and t_1_. (**b**) p_v_ and t_2_. (**c**) t_1_ and t_2_.

**Figure 4 foods-11-02458-f004:**
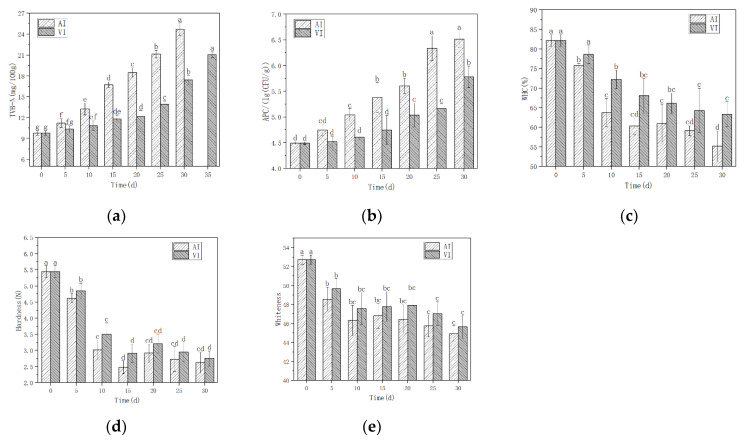
Graph of quality parameter change compared between VI and AI with combined preservative during storage at −2 °C. (**a**) TVB-N values; (**b**) aerobic plate count values (APC); (**c**) water-holding capacity values (WHC); (**d**) hardness; (**e**) whiteness. Note: Different lowercase letters indicate significant differences between groups with different storage storage days (*p* < 0.05).

**Table 1 foods-11-02458-t001:** References for studying aquatic products and meat using VI technology involving parameters of p_v_, t_1_, and t_2_.

Key Words	References	VI Parameter			Preserving Temperature (°C)
		p_v_ (kPa)	t_1_ (min)	t_2_ (min)	
Aquatic products; preservatives; VI	X. Zhao et al. [23,27,28]	5	15	10	4
	Y. zhang et al. [24]	70	15	Not mentioned	Not mentioned
	S. Khursheed et al. [25,26]	5	15/7.5	15/7.5	4
	A. Andres-bello et al. [30]	5	5	5	4
	L.M.Goeller et al. [31]	0.35	5	25	3
Meat; aquatic products; salting; VI	F.Deumier et al. [32]	15	1.67	0.333	7
	A. Elif [5]	25	10	Not mentioned	28
	H.Demir et al. [21]	100	40	Not mentioned	23
	A.Tomac et al. [18]	10	5	240	4
	N.Ramirez et al. [22]	15	4	Not mentioned	6
	C.Figueroa et al. [33]	94.66	5	2	8
	M.G.Martins et al. [4]	91	5	10	30
	M.J.Larrazabal et al. [34]	5	15	75	5
	M.Bampi et al. [35]	94.66	20	340	10

**Table 2 foods-11-02458-t002:** Coded values of the treatment variables for all VI solutes employed.

Parameters	Coded Symbol	Coded Levels		
		−1	0	1
p_v_ (kPa)	A	40	60	80
t_1_ (min)	B	15	20	25
t_2_ (min)	C	6	9	12

**Table 3 foods-11-02458-t003:** Box–Behnken design with the independent variables p_v_, t_1_, and t_2_ in their original and coded forms and results for APC, TVB-N, and CS.

Run No.	Code			Responses		
	p_v_ (kPa)	t_1_ (min)	t_2_ (min)	APC (lg CFU/g)	TVB-N (mg/100 g)	CS
1	60 (0)	15 (−1)	6 (−1)	5.52	16.69	0.373
2	40 (−1)	20 (0)	6 (−1)	5.63	17.55	0.180
3	40 (−1)	20 (0)	6 (−1)	5.38	15.96	0.578
4	60 (0)	25 (1)	6 (−1)	5.41	16.07	0.539
5	40 (−1)	15 (−1)	9 (0)	5.73	17.98	0.044
6	80 (1)	15 (−1)	9 (0)	5.67	18.22	0.069
7	60 (0)	20 (0)	9 (0)	5.27	14.75	0.810
8	60 (0)	20 (0)	9 (0)	5.33	14.36	0.801
9	60 (0)	20 (0)	9 (0)	5.22	15.23	0.800
10	60 (0)	20 (0)	9 (0)	5.21	14.44	0.898
11	60 (0)	20 (0)	9 (0)	5.16	13.92	1.000
12	40 (−1)	25 (1)	9 (0)	5.44	16.11	0.508
13	80 (1)	25 (1)	9 (0)	5.19	14.03	0.961
14	60 (0)	15 (−1)	12 (1)	5.71	18.37	0.018
15	40 (−1)	20 (0)	12 (1)	5.57	17.72	0.213
16	80 (1)	20 (0)	12 (1)	5.51	17.35	0.308
17	60 (0)	25 (1)	12 (1)	5.32	16.36	0.585

Note: Values between parentheses are the coded forms of the variables.

**Table 4 foods-11-02458-t004:** Changes of WGR, APC and TVB-N of tilapia fillets under different p_v_ with fixed t_1_ and t_2_.

	Storage Time (d)	p_v_				
		Atmosphere	20 kPa	40 kPa	60 kPa	80 kPa
WGR (%)		0.908 ± 0.146 ^d^	1.160 ± 0.061 ^c^	1.378 ± 0.120 ^b^	1.661 ± 0.081 ^a^	1.707 ± 0.150 ^a^
APC lg(CFU/g)	0	4.47 ± 0.02 ^Ad^	4.47 ± 0.02 ^Ae^	4.47 ± 0.02 ^Ad^	4.47 ± 0.02 ^Ac^	4.47 ± 0.02 ^Ae^
	5	4.86 ± 0.16 ^Acd^	4.68 ± 0.08 ^Ad^	4.60 ± 0.10 ^Acd^	4.60 ± 0.23 ^Abc^	4.66 ± 0.05 ^Ad^
	10	5.15 ± 0.19 ^Abc^	4.91 ± 0.02 ^Ac^	4.89 ± 0.42 ^Abc^	4.73 ± 0.09 ^Ab^	4.82 ± 0.13 ^Ac^
	15	5.43 ± 0.15 ^Ab^	5.31 ± 0.05 ^ABb^	5.22 ± 0.09 ^BCab^	5.09 ± 0.02 ^Ca^	5.12 ± 0.03 ^Cb^
	20	5.83 ± 0.40 ^Aa^	5.72 ± 0.07 ^Aa^	5.50 ± 0.01 ^ABa^	5.26 ± 0.05 ^Ba^	5.36 ± 0.06 ^Ba^
TVB-N (mg/100 g)	0	9.78 ± 0.25 ^Ae^	9.78 ± 0.25 ^Ae^	9.78 ± 0.25 ^Ae^	9.78 ± 0.25 ^Ae^	9.78 ± 0.25 ^Ac^
	5	13.42 ± 0.19 ^Ad^	12.66 ± 0.62 ^ABd^	12.38 ± 0.49 ^ABd^	12.07 ± 0.43 ^ABd^	11.82 ± 1.38 ^Bb^
	10	15.62 ± 0.37 ^Ac^	13.54 ± 0.26 ^Bc^	13.24 ± 0.29 ^BCc^	12.81 ± 0.39 ^CDc^	12.55 ± 0.38 ^Db^
	15	16.81 ± 0.47 ^Ab^	15.78 ± 0.61 ^Bb^	15.46 ± 0.24 ^Bb^	14.95 ± 0.21 ^Bb^	15.32 ± 0.47 ^Db^
	20	18.47 ± 0.49 ^Aa^	16.97 ± 0.38 ^Ba^	16.35 ± 0.25 ^BCa^	15.84 ± 0.18 ^Ca^	16.03 ± 0.34 ^Ca^

Note: Different lowercase letters indicate significant differences between groups with different vacuum pressures for the same storage days (*p* < 0.05), and different capital letters indicate significant differences between groups with different storage days under the same pressure (*p* < 0.05).

**Table 5 foods-11-02458-t005:** Changes in WGR, APC, and TVB-N of tilapia fillets under different t_1_ with fixed p_v_ and t_2_.

	Storage Time (d)	t_1_				
		5 min	10 min	15 min	20 min	25 min
WGR (%)		1.319 ± 0.134 ^c^	1.463 ± 0.132 ^bc^	1.607 ± 0.120 ^ab^	1.783 ± 0.063 ^a^	1.665 ± 0.067 ^ab^
APC lg(CFU/g)	0	4.50 ± 0.03 ^Ae^	4.50 ± 0.03 ^Ae^	4.50 ± 0.03 ^Ac^	4.50 ± 0.03 ^Ac^	4.50 ± 0.03 ^Ad^
	5	4.82 ± 0.03 ^Ad^	4.67 ± 0.09 ^Ad^	4.60 ± 0.07 ^Ac^	4.58 ± 0.02 ^Ac^	4.70 ± 0.13 ^ABc^
	10	5.18 ± 0.05 ^Ac^	5.08 ± 0.04 ^ABc^	4.99 ± 0.08 ^BCb^	4.95 ± 0.06 ^Cb^	5.11 ± 0.01 ^Ab^
	15	5.44 ± 0.06 ^Ab^	5.37 ± 0.03 ^ABb^	5.28 ± 0.07 ^BCa^	5.19 ± 0.06 ^Ca^	5.30 ± 0.06 ^Ba^
	20	5.60 ± 0.08 ^Aa^	5.47 ± 0.05 ^ABa^	5.39 ± 0.10 ^BCa^	5.25 ± 0.08 ^Ca^	5.37 ± 0.03 ^BCa^
TVB-N (mg/100 g)	0	9.93 ± 0.09 ^Ae^	9.93 ± 0.09 ^Ad^	9.93 ± 0.09 ^Ad^	9.93 ± 0.09 ^Ae^	9.93 ± 0.09 ^Ae^
	5	13.21 ± 0.43 ^Ad^	12.74 ± 0.75 ^Ac^	11.76 ± 0.30 ^Bc^	11.53 ± 0.31 ^Bd^	12.32 ± 0.42 ^ABd^
	10	15.33 ± 0.17 ^Ac^	14.97 ± 0.42 ^ABb^	14.52 ± 0.41 ^BCb^	13.91 ± 0.30 ^Dc^	14.30 ± 0.20 ^CDc^
	15	16.59 ± 0.19 ^Ab^	15.62 ± 0.11 ^Bb^	15.67 ± 0.29 ^Ba^	14.73 ± 0.27 ^Db^	15.14 ± 0.18 ^Cb^
	20	17.32 ± 0.22 ^Aa^	17.13 ± 0.28 ^Aa^	16.18 ± 0.56 ^BCa^	15.83 ± 0.23 ^Ca^	16.49 ± 0.32 ^Ba^

Note: Different lowercase letters indicate significant differences between groups with different vacuum pressures time (t_1_) for the same storage days (*p* < 0.05), and different capital letters indicate significant differences between groups with different storage days under the same t_1_ (*p* < 0.05).

**Table 6 foods-11-02458-t006:** Changes in WGR, APC, and TVB-N of tilapia fillets under different t_2_ with fixed p_v_ and t_1_.

	Storage Time/d	t_2_				
		3 min	6 min	9 min	12 min	15 min
WGR (%)		1.441 ± 0.081 ^a^	1.536 ± 0.142 ^a^	1.642 ± 0.127 ^a^	1.575 ± 0.057 ^a^	1.608 ± 0.110 ^a^
APC lg(CFU/g)	0	4.48 ± 0.04 ^Ae^	4.48 ± 0.04 ^Ae^	4.48 ± 0.04 ^Ae^	4.48 ± 0.04 ^Ae^	4.48 ± 0.04 ^Ae^
	5	4.79 ± 0.05 ^Ad^	4.61 ± 0.09 ^Bd^	4.63 ± 0.05 ^Bd^	4.72 ± 0.05 ^ABd^	4.79 ± 0.04 ^Ad^
	10	5.10 ± 0.06 ^Ac^	5.04 ± 0.04 ^ABc^	4.96 ± 0.04 ^Bc^	4.95 ± 0.05 ^Bc^	4.99 ± 0.05 ^Bc^
	15	5.34 ± 0.02 ^Ab^	5.29 ± 0.04 ^ABb^	5.21 ± 0.03 ^Cb^	5.23 ± 0.03 ^Cb^	5.28 ± 0.04 ^BCb^
	20	5.56 ± 0.03 ^Aa^	5.45 ± 0.05 ^ABa^	5.39 ± 0.04 ^Ba^	5.39 ± 0.11 ^Ba^	5.49 ± 0.06 ^ABa^
TVB-N (mg/100 g)	0	9.85 ± 0.11 ^Ae^	9.85 ± 0.11 ^Ae^	9.85 ± 0.11 ^Ae^	9.85 ± 0.11 ^Ad^	9.85 ± 0.11 ^Ae^
	5	13.47 ± 0.66 ^Ad^	12.61 ± 0.40 ^Ad^	12.78 ± 0.47 ^Ad^	12.67 ± 0.38 ^Ac^	13.06 ± 0.25 ^Ad^
	10	14.95 ± 0.10 ^Ac^	14.29 ± 0.29 ^Bc^	13.47 ± 0.35 ^Cc^	13.35 ± 0.45 ^Cc^	14.32 ±0.36 ^Bc^
	15	15.87 ± 0.33 ^Ab^	15.27 ± 0.27 ^ABb^	14.45 ± 0.39 ^Cb^	14.51 ± 0.43 ^Cb^	14.86 ± 0.26 ^BCb^
	20	16.89 ± 0.34 ^Aa^	16.28 ± 0.35 ^ABa^	15.71 ± 0.43 ^Ba^	15.68 ± 0.50 ^Ba^	16.05 ± 0.27 ^Ba^

Note: Different lowercase letters indicate significant differences between groups with different atmosphere restore time (t_2_) for the same storage days (*p* < 0.05), and different capital letters indicate significant differences between groups with different storage days under the same t_2_ (*p* < 0.05).

**Table 7 foods-11-02458-t007:** Analysis of variance for the parameter estimations obtained from multiple linear regression and for the lack-of-fit test for p_v_, t_1_, and t_2_.

Factor	Aerobic Plate Counts (APC)											
	SS	DF	F	*p*	SS	DF	F	*p*	SS	DF	F	*p*
Model	0.548	9	16.45	0.0006 ***	34.75	9	11.08	0.0022 **	1.69	9	16.52	0.0006 ***
A (p_v_)	0.048	1	12.98	0.0087 **	1.81	1	5.18	0.0570	0.12	1	10.34	0.0147 *
B (t_1_)	0.202	1	54.48	0.0002 ***	9.44	1	27.08	0.0012 **	0.55	1	47.85	0.0002 ***
C (t_2_)	0.004	1	0.98	0.3561	1.56	1	4.47	0.0724	0.04	1	3.27	0.1136
AB	0.009	1	2.44	0.1623	1.35	1	3.86	0.0902	0.05	1	4.02	0.0851
AC	0.009	1	2.44	0.1623	0.37	1	1.07	0.3359	0.02	1	2.01	0.1989
BC	0.020	1	5.30	0.0549	0.48	1	1.39	0.2776	0.04	1	3.53	0.1025
A^2^	0.096	1	25.94	0.0014 **	5.65	1	16.22	0.0050 **	0.29	1	25.48	0.0015 **
B^2^	0.059	1	15.98	0.0052 **	3.31	1	9.49	0.0178 *	0.17	1	15.32	0.0058 **
C^2^	0.075	1	20.28	0.0028 **	8.81	1	25.26	0.0015 **	0.33	1	28.83	0.0010 **
Lack of Fit	0.009	3	0.74	0.5822	1.49	3	2.10	0.2426	0.05	3	2.14	0.2376
Pure Error	0.017	4			0.95	4			0.03	4		
R^2^	0.955				0.934				0.955			
Adeq Precisior	11.065								11.428			

Note: SS Sum of squares, DF Degree of freedom. * Significant at *p* < 0.05; ** Significant at *p* < 0.01; *** Significant at *p* < 0.001.

**Table 8 foods-11-02458-t008:** Regression coefficients of responses in terms of actual factors that are specified in the original units for each factor.

	APC (lg CFU/g)	TVB-N (mg/100 g)	CS
intercept a_0_	9.5158	48.2263	−6.6737
Linear			
a_1_	−0.0468 **	−0.3011	0.0748 *
a_2_	−0.1509 ***	−1.0788 **	0.2537 ***
a_3_	−0.2141	−2.5871	0.4781
interaction			
a_12_	−0.0005	−0.0058	0.0011
a_13_	0.0008	0.0051	−0.0013
a_23_	−0.0047	−0.0232	0.0067
Quadratic			
a_11_	0.0004 **	0.0029 **	−0.0007 **
a_22_	0.0047 **	0.0355 *	−0.0081 **
a_33_	0.0148 **	0.1607 **	−0.0310 **

Note: * Significant at *p* < 0.05; ** Significant at *p* < 0.01; *** Significant at *p* < 0.001.

## Data Availability

Data are contained within the article.

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
