# Peer review of "Vacuum Impregnation Process Optimization for Tilapia with Biopreservatives at Ice Temperature"

_foods, 2022, doi:10.3390/foods11162458_

Round 1
Reviewer 1 Report
The manuscript needs extensive revision
Some non-English references can be replaced by English ones
Colored figures rather than gray and white can be used
Some specific comments:
Line 13: Please replace “the ideal” by “processing conditions including”
Line 16: “base” by “basic” Pleas do the same throughout the whole manuscript
Lines 35-36: Replace “Ice-temperature storage is a regularly used method” by “Ice-temperature storage and freezing are regularly used methods“
Line 38: replace “superior than those of frozen storage, consumers are more receptive to the items“ by “ in fresh samples than those of frozen storage, consumers prefer fresh products“”
Line 38: reference should be provided here. The following references for example can be used:
-Emerging techniques for differentiation of fresh and frozen–thawed seafoods: Highlighting the potential of spectroscopic techniques, Molecules 25 (19), 4472
- Exploring the Potential of Fluorescence Spectroscopy for the Discrimination between Fresh and Frozen-Thawed Muscle Foods, Photochem 1 (2), 247-263
Line 335: replace “TVB-N value(TVB-N)” by “TVB-N values”
Lines 366, 388, etc. verify numbering
Reviewer 2 Report
The paper entilted « Vacuum impregnation process optimization for tilapia with biopreservatives at ice temperature » is on the optimization of a process to enhance the penetration of additives in fish muscle to improve fish quality and to increase the storage time. This study is interesting because with the optimization it completes the studies of Zhao et al. on tilapia too.
But, this paper cannot be published as it is. It isn’t clearly written and the English language is very poor. The paper has to be improved a lot all along the manuscript.
Here are my comments to the authors to help them make corrections of the article. English not being my native language, so I will not correct this one.
- Lines 67 to 69: this is particularly no understandable what are the “proteins and lipids used in freeze (-18°C) thaw (-4°C) cycles”?
- Lines 70-72: No, Shiekh Khursheed et al. combined pulsed electric field, chamuang leaf extract and cold plasma to preserve Litopenaeus vannamei.
- Lines 76-77: it is unclear, you have to precise the study of Min et al.
- Lines 80-81: Could you precise, it’s unclear. Different from what?
- Line 92: you have to explain RSM as you wrote it for the first time in the manuscript (summary excluded).
- Line 98: Scientific name must be in italics
- Line 122 and all through the manuscript: what is the preservative liquid? You never indicated the mixture you used. You talked about “biopreservatives” but we don’t know what are your preservative(s). The sentence is not a sentence to be written in a paper.
- Line 137: Kjeldahl analyser is an equipment, not a method.
- Line 184: how do you choose your “pertinent” variables (WGR, APC and TVB-N)?
- Line 236: no, this is the influence of pressure parameters on the variables (WGR, APC, TVB-N), and not the opposite.
- Lines 246-247: “Because the biopreservative liquid has completely filled the fish cell spacing and reached a saturated state at 60kPa, the internal and external pressures have reached equilibrium, and the biopreservative can no longer penetrate into the fish tissue”. How you can have this conclusion? Do you have also made microscopy examination?
- Line 265: same comment as for the line 236. This is the opposite. This is the influence on time on the variables.
- Lines 278-280: this sentence is particularly not understandable “This could because the fish fillets acqured a low-pressure during in cell structure during the t1 stage and the system pressure restored to atmospheric pressure during the t2 stage.” Please, clarify.
- Lines 299-302: this sentence has also to be clarified: “On all response parameters, the linear effect of t2 and all the interaction variables were determined to be nonsignificant on any response parameters, however significant quadratic relationships were seen with APC and TVB-N .”
- Lines 376-377: unclear too: “the fish fillets began to spoil (so other quality-related parameters only detected for 30 days)”
- Line 383: what is “the composite biopreservative”?
- Lines 385 to 387: unclear, to be specified.
- Use “degradation” instead of “decomposition” for the protein gel structure.
- Line 493: “on quality” and not “of quality”.
- References: they must be carefully checked without, for example, forget italics to write the scientific names .
Round 2
Reviewer 1 Report
The authors addressed my concerns.